# The Profile of Polyphenolic Compounds, Contents of Total Phenolics and Flavonoids, and Antioxidant and Antimicrobial Properties of Bee Products

**DOI:** 10.3390/molecules27041301

**Published:** 2022-02-15

**Authors:** Tomasz Sawicki, Małgorzata Starowicz, Lucyna Kłębukowska, Paweł Hanus

**Affiliations:** 1Department of Human Nutrition, Faculty of Food Sciences, University of Warmia and Mazury in Olsztyn, 45f Słoneczna Street, 10-718 Olsztyn, Poland; 2Department of Chemistry and Biodynamics of Food, Institute of Animal Reproduction and Food Research of Polish Academy of Sciences, 10 Tuwima Street, 10-748 Olsztyn, Poland; m.starowicz@pan.olsztyn.pl; 3Department of Industrial and Food Microbiology, Faculty of Food Sciences, University of Warmia and Mazury in Olsztyn, Plac Cieszyński 1, 10-726 Olsztyn, Poland; lutkak@uwm.edu.pl; 4Department of Technology and Plant Product Quality Assessment, University of Rzeszów, 4 Zelwerowicza Street, 35-601 Rzeszów, Poland; phanus@ur.edu.pl

**Keywords:** antimicrobial activity, antioxidant capacity, bee products, honey, phenolics, flavonoids

## Abstract

This study aimed to characterize bee products (bee bread, bee pollen, beeswax, and multiflorous honey) with the profile of phenolic compounds, total phenolic (TPC) and flavonoid (TFC) contents, and antioxidant and microbiological properties. The TP and TF contents could be ordered as follows: bee pollen > bee bread > beeswax > honey. The UPLC−PDA−MS/MS analysis allowed identifying 20 polyphenols. Sinapic acid dominated in bee pollen, gallic acid in the bee bread and honey, while pinobanksin was the major compound of beeswax. The data showed that bee pollen and bee bread had a stronger antioxidant potential than honey and beeswax. Moreover, the antibacterial activity of the bee products was studied using 14 bacterial strains. Bee bread’s and bee pollen’s antimicrobial activity was higher towards Gram-negative strains. In comparison, honey was more potent in inhibiting Gram-positive bacteria. Our study indicates that bee products may represent valuable sources of bioactive compounds offering functional properties.

## 1. Introduction

Numerous studies have pointed out the high nutritional value and health benefits of such bee products as pollen, bee bread, honey, and beeswax [1,2,3]. It has been proven that their positive impact on the human body (antioxidant, anti-microbial, anti-fungal, anti-inflammatory, etc.) could be related to the high content of specific bioactive compounds. Baltršaitytė et al. [4] determined the phenolic composition of Lithuanian honey samples and showed they were rich in *p*-coumaric acid, kaempferol, chrysin, and apigenin. They found a strong correlation between the content of phenolic compounds and their antioxidant activity. Also, Socha et al. [5] found a linear correlation between phenolics content and antioxidant activity in different varieties of Polish honey. Flavonol glycosides of quercetin, isorhamnetin, and kaempferol were determined as major components of the bee pollen [6]. In another study, flavonoid glycosides of 6-methoxyquercetin (patuletin) were also detected in Brazilian pollen samples, together with hydroxycinnamic acid amide derivatives [7].

The antioxidant activity of different bee products has been already measured using different in vitro methods, including 1,1-diphenyl-2-picrylhydrazyl (DPPH), 2,2′-azino-bis(3-ethylbenzothiazoline-6-sulfonic acid) (ABTS), and photochemiluminescence (PCL) assays, and reducing activity measured by cupric (CUPRAC) and ferric reducing antioxidant power (FRAP) [8,9,10,11]. The strongest antioxidant activity was found for propolis and pollen, followed by honey [9]. Besides antioxidant activity, bee products offer a very high antimicrobial potential [12,13]. However, all the bioactive properties of bee products vary as affected by the geographical/regional origin, climate changes, and/or cultivation season. According to our previous research, the TPC and antioxidant capacity might also be related to the producer [10].

In the case of honey, the above-mentioned properties depend on their origin and variety as well as harvest season and conditions [14,15]. Available literature shows that the dark-type honeys elicit more health benefits than the light-type ones. Moreover, current studies indicate that multifloral honeys are healthier than monofloral honeys [15]. Honey is the most popular bee product, while bee pollen and bee bread are increasingly consumed as functional food, or superfood, due to their high concentration of bioactive substances [16]. In the case of beeswax, it is widely used in the cosmetic industry and also in the food industry to protect ripened cheeses [17]. Considering the above, it is very important to know the composition and biological activity of bee products that are intended to be used, or are already used, as dietary supplements. This is the first study in the literature to evaluate and compare four different bee products collected from the same hives. Thus, it aimed to characterize the bee products (beeswax, bee bread, bee pollen, and honey) by the profile of phenolic compounds, TPC and TFC, antioxidant activity, and microbiological properties.

## 2. Results and Discussion

### 2.1. Total Phenolic Content (TPC) and Total Flavonoid Content (TFC)

The TPC and TFC in bee bread, bee pollen, beeswax, and honey are presented in Figure 1. The TPC was within the range from 0.47 ± 0.04 to 32.52 ± 2.19 mg GAE/g, and its highest value was determined in the bee pollen (32.52 ± 2.19 mg GAE/g). Almost four times lower TPC value was determined in the bee bread (8.23 ± 0.24 mg GAE/g). Significantly (*p* < 0.05) lower TPC values were detected in beeswax and honey, being approximately 46 and 69 times lesser, respectively. In comparison, bee pollen from Italy was characterized by a lower concentration of TPC, ranging from 13.53 ± 0.40 to 24.75 ± 0.78 mg GAE/g depending on the botanical origin of samples [18]. Another Italian research also showed lower TPC values (4.20 ± 0.40–29.60 ± 0.90 mg GAE/g) in bee pollen collected in 2014, and 2015 [19]. Lower TPC values were also detected in the bee pollen collected in the south of Poland (27.03 mg GAE/g) [20]. In the case of bee bread, studies have reported that the TPC of its samples collected in different regions varies from 2.5 to 37.15 mg GAE/g [14,21]. The results obtained in our study for bee bread (8.23 ± 0.24 mg GAE/g) are within this range. Furthermore, according to our recently published results, this bee product obtained from the northeast part of Poland had a lower concentration of phenolic compounds (by 41%) [10] compared to the bee bread from the central part of Poland (this study). The total phenolic content of honey was similar to the data reported by Socha et al. [22] for multifloral honey, i.e., TPC between 0.42 and 0.56 mg GAE/g. Also, similar values to those of the multifloral honey (0.47 ± 0.04 mg GAE/g) obtained from the central part of Poland and analyzed in the present study were found in the multifloral honey from the northeastern Poland (0.49 ± 0.02 mg GAE/g) [10]. In turn, Wesołowska and Dżugan [11] determined almost 2-times lower values of TPC in multifloral honey obtained from the southern Poland, compared to our study. To the best of our knowledge, there is a lack of data on the content of bioactive compounds in beeswax. Our data of TPC are higher than the values reported for Spanish beeswax [23]. However, the other papers showed only the content of polyphenols in the by-products from beeswax recycling process [24].

A similar tendency was observed in the case of the TFC values, which oscillated between 0.07 ± 0.00 and 11.77 ± 0.15 mg QE/g. The highest TFC value was found for bee pollen (11.77 ± 0.15 mg QE/g), while the lowest one for honey (0.07 ± 0.00 mg QE/g). The TF content in the bee pollen was approximately 60, 96, and 99% higher than in the bee bread, beeswax, and honey, respectively. The previously published data showed an ambiguous concentration of flavonoids in bee pollen. Rzepecka-Stojko et al. [20] determined a higher level of TFC (20.22 mg QE/g) in the Polish bee pollen, whereas Mayda et al. [14] found only 2.62–4.44 mg QE/g of flavonoids in the Turkish bee pollen. In the case of bee bread’s flavonoids, the Turkish bee bread was characterized by a lower level of these compounds (1.81–3.74 mg QE/g), than the Polish bee bread [14]. However, Zuluga et al. [25] determined a similar TFC in the Colombian bee bread (1.9–4.5 mg QE/g). With regard to the next bee product, the previously published data showed a similar TFC in the Polish multifloral honeys (0.05–0.14 mg QE/g) [22] with those obtained in our study. On the other hand, other multifloral honeys also originating from Poland, had more than 2-times lower level of flavonoids [26]. In the case of beeswax, previously published data indicated a much lower content of flavonoids in beeswax samples from Spain [26]. In addition, flavonoids account for 25 to 75% of the total polyphenols (TP) in bee products [19]. In our study, the contribution of flavonoids in honey was lower, and constituted about 13% of TP. The difference in the content of bioactive compounds in honey may be due to their origin [14,19]. However, in the case of bee bread, bee pollen, and beeswax, the contribution of TFC was higher than in honey and was approximately 36, 27, and 42% of TP, respectively. The obtained data indicate that the bee products represent good sources of phenolic and flavonoid compounds. The order of polyphenols content in the bee products was as follows: bee pollen > bee bread > beeswax > honey. To summarize, the various contents of phenolic and flavonoid compounds detected in bee products may result from the three main factors: method of extraction, botanical origin, and location of harvest [10,14,18,19]. Mayda et al. [14] demonstrated that TP/TF content varied significantly in the samples of bee bread and bee pollen from the same beehive located in different regions. The range of TPC values for bee bread was from 26.69 to 43.42 (mg GAE/g), and for bee pollen it was from 8.26 to 12.71 (mg GAE/g) [14]. The same tendency was noted for TFC, which varied between 2.62-4.44 mg QE/g in bee-pollen samples. In bee bread, TFC values were not stable either and ranged from 1.81 to 3.74 mg QE/g. Rocchetti et al. [19] determined the highest TPC in bee pollen with the predominant *Daucus* and *Coriandrum* pollen, whereas the lowest one in the samples with preponderance of *Magnolia* pollen.

### 2.2. Polyphenolic Profile in the Bee Products

The next stage of this study entailed determinations of the profile and contents of the main phytochemicals in the analyzed bee products. The characterization of the polyphenolic profile was carried out mainly to determine which specific compounds might have the strongest effect upon the antioxidant and anti-microbiological properties of the bee products. Polyphenolics were determined by ultra-performance liquid chromatography coupled with a mass spectrometer (UPLC−PDA−MS/MS), and respective results are presented in Table 1. Twenty compounds were identified in bee products, eleven of which were flavonoids (sakuranetin dimer, rutin, isorhamnetine 3-*O*-rutinoside, quercetin 3-*O*-glucuronide, orientin, vitexin, quercetin, epicatechin, kaempferol, pinobanksin, and apigenin), eight represented phenolic acids (gallic, neochlorogenic, chlorogenic, protocatechuic, caffeic, sinapic, 3,4-di-*O*-caffeoylquinic, and protocatechuic acid-*O*-hexoside acids), and one compound belonged to ellagitannins (ellagic acid). Previously published data have also shown the presence of three groups of polyphenolic compounds in honey, bee pollen, and bee bread [27].

Each bee product had its unique polyphenol profile, which is shown in Table 2. Gallic, ellagic, neochlorogenic, chlorogenic, protocatechuic, and sinapic acids were detected in honey, bee pollen, and bee bread. Rutin, isorhamnetin 3-*O*-rutinoside, quercetin 3-*O*-glucuronide, and epicatechin were found in bee bread and bee pollen. In addition, protocatechuic acid-*O*-hexoside and pinobanksin were present in the bee bread, bee pollen, and beeswax. However, some compounds were present only in particular bee products. Sakuranetin dimer, caffeic acid, and quercetin were detected only in honey, while orientin and vitexin only in bee bread. Moreover, three compounds (3,4-di-*O*-caffeoylquinic acid, kaempferol, and apigenin) were found only in beeswax.

Habryka et al. [26] found six phenolic acids (ferulic, gallic, *p*-hydroxybenzoic, caffeic, *p*-coumaric, and protocatechuic acids) and four flavonoids (kaempferol, chrysin, galangin, and quercetin) in Polish multifloral honey. Results of the study conducted by Rzepecka-Stojko et al. [20] showed the presence of six phenolic acids (gallic, caffeic, ferulic, 4-hydroxycinnamic, 4-*t*-*p*-coumaric, and *t*-cinnamic acids) and five flavonoids (rutin, myrycithin, quercetin, kaempferol, and isorhamnetin) in bee pollen from the southern part of Poland. In the cited study, the bee pollen did not contain nine compounds (ellagic, neochlorogenic, chlorogenic, protocatechuic, sinapic, and protocatechuic acid-*O*-hexoside acids, as well as quercetin 3-*O*-glucuronide, epicatechin, and pinobanksin) detected in the samples examined in our study. In the case of bee bread, the Moroccan’s bee bread sample was found to contain thirteen phenolic compounds, mainly flavonols glycoside derivatives, especially quercetin, kaempferol, isorhamnetin, and methylherbacetrin derivatives [28]. In contrast, Isidorov et al. [29] identified using the GC-MS analysis, naringenin, kaempferol, apigenin, isorhamnetin, and quercetin in bee bread from Latvia, Russia, and Poland. In turn, ten phenolic compounds (mainly flavonoids derivatives) have been identified in the beeswax by-products [24].

The highest total content of phenolic acids, expressed as a sum of individual compounds, was detected in bee pollen (52.51 µg/g), followed by bee bread (35.91 µg/g), honey (1.39 µg/g), and beeswax (0.02 µg/g). The main compound among the phenolic acids in the bee pollen was sinapic acid, followed by gallic acid (42.8% and 21.3% of the sum of individual polyphenolic compounds, respectively; Table 2). The available literature data shows that the major phenolic acid in bee pollen was gallic acid [20], and that gallic acid was found to be the dominant phenolic acid in bee bread and honey (32.6 and 69.2%, respectively). Furthermore, sinapic acid was proved to be the second most abundant compound of bee bread (27.3%), whereas chlorogenic acid to be the second most dominant phenolic acid of honey (13.2%). This was an important finding because sinapic, gallic, and chlorogenic acids are indicators of the antimicrobial and antioxidant activities [30,31]. In the case of beeswax, the major phenolic acid was protocatechuic acid-*O*-hexoside (3.4%).

In the case of flavonoids, the highest TFC presented as a sum of individual flavonoid compounds was found in bee pollen (25.18 µg/g), followed by bee bread (18.87 µg/g), then beeswax (0.37 µg/g), and honey (0.11 µg/g). The presented data agree with the results obtained using the spectrophotometric method presented above. The major flavonoids of bee pollen were rutin (10% of the TFC; Table 2), followed by quercetin 3-*O*-glucuronide and epicatechin (7.9 and 7.2%, respectively). These results are consistent with findings reported by Rzepecka-Stojko et al. [20], who identified rutin as the major flavonoid of bee pollen. In the case of bee bread, the dominant flavonoid was vitexin, which accounted for 15.2% of TFC, and was followed by rutin accounting for 5.1% of TFC. In comparison, Bayram et al. [32] reported that rutin followed by quercetin were the main flavonoids of the Turkish bee bread. Moreover, these authors did not find vitexin in bee bread, while in our research, quercetin was only present in honey The major flavonoid compound of honey turned out to be sakuranetin dimer which accounted for 4.6% of the TFC. In the case of beeswax, the main flavonoid was pinobanksin which constituent 85.7% of the total amount of flavonoids.

The dominant bioactive compounds of bee pollen, bee bread, and honey were phenolic acids, which constituted 66.1%, 63.8%, and 86.6% of the sum of individual polyphenolic compounds. On the other hand, flavonoids were the main group of compounds found in beeswax (94.8%). Moreover, bee bread shared a common polyphenolic profile with that of the other samples (bee pollen and honey), likely since it is a combination of honey and pollen [3,10]. Furthermore, our study indicated that the polyphenols present in bee pollen were more significant contributors to the profile of these compounds of bee bread than these present in honey.

Differences in the polyphenolic profiles and contents can be related to the method of extraction and a less sensitive analytical technique. Moreover, the compound number detected may be related to the region of bee products origin. The previous studies have also shown a relationship between the profile of volatile compounds in bee products and their area of origin [10,32].

### 2.3. Antioxidant Activity and Reducing Potential of Bee Products

The antioxidant activity of the obtained extracts of bee products was measured by the ABTS and DPPH assays and the PCL method, which included two different approaches (hydrophilic and lipophilic conditions). The tested bee products were characterized by other antioxidant activities (Table 3). The order of average antioxidant activity for the bee products was as follows: ABTS > ACL (lipophilic antioxidants) > ACW (hydrophilic antioxidants) > DPPH. Moreover, the antioxidant activity determined by the ABTS assay was highly correlated with the data evaluated by the ACW (r = 0.933) and DPPH (r = 0.900) methods, and moderately correlated with the ACL values (r = 0.685). The antioxidant activity values determined by the DPPH test were highly correlated with the ACW data (r = 0.874), and moderately with the ACL values (r = 0.567). In addition, the values obtained in the ACW assay were highly correlated with the data measured using the ACL test (r = 0.882).

According to the obtained data, bee bread, bee pollen, and beeswax had higher ACL values than ACW, while the importance of ACW was higher only in the honey (Table 3). Bee bread had more than 6-times higher ACL values than the ACW, while the bee pollen had 3-times higher ACL/ACW contribution. However, the highest ACL/ACW contribution was detected in beeswax, and the ACL value was more than 40-times higher than ACW. The ACL values were in the following order: bee bread > bee pollen > beeswax > honey, whereas ACW values could be ordered as follows: bee bread > bee pollen > beeswax = honey. Bee bread and multifloral honey were also tested by Sawicki et al. [10] for the ability to scavenge superoxide anion radicals (O^2−^). The data showed that bee bread’s extracts have a high antioxidative ability due to the content of lipophilic antioxidants (ACL), but the hydrophilic antioxidants (ACW) of bee bread have significantly lower antioxidant status. In the case of honey, a higher antioxidative ability of hydrophilic antioxidants had been noticed in comparison to the honey’s lipophilic antioxidants. Moreover, the results for honey are consistent with the findings from other studies examining multifloral Polish honey [11].

The results of the DPPH test showed that the bee pollen possessed the highest antioxidant activity among the examined bee products (*p* < 0.05), which may, presumably, be due to the large number of bioactive compounds detected in bee pollen. Furthermore, such compounds as sinapic acid, gallic acid, and rutin were characterized by high antioxidant activity dominated in this product [30,31]. The antioxidant activity of bee bread, beeswax, and honey were approximately 40, 97, and 99% lower than the antioxidant activity determined by the DPPH method for bee pollen. Our results for the bee pollen are consistent with the data obtained by Rocchetti et al. [19], who determined the DPPH values for bee pollen of various origin to range from 11.9 to 134.7 µmol Trolox/g. The obtained multifloral honey DPPH values are slightly lower than those reported by Habryka et al. [26]. Moreover, our data obtained for beeswax was about 3-times lower than those obtained for the Spanish beeswax [23]. However, DPPH values obtained for bee bread, multifloral honey, and bee pollen were difficult to compare with previously published data mainly due to different units of measure adopted [14,20,21,28].

In the ABTS assay case, the highest antioxidant activity value was also obtained for the bee pollen (32.56 ± 0.30 mmol Trolox/g). In addition, the bee bread was found to have an equally high ability to scavenge ABTS radicals (31.60 ± 0.16 mmol Trolox/g). The differences between these two bee products were statistically significant. Two-times lower value of antioxidant activity determined in the ABTS assay was obtained for the extracts of honey compared to the bee pollen. On the other hand, the lowest antioxidant activity values examined using the ABTS test were obtained for beeswax (5.96 ± 0.05 mmol Trolox/g).

The highest reducing potential determined in the FRAP test was obtained for the bee pollen (76.94 ± 4.48 µmol Trolox/g; Table 3). The bee bread and honey also featured high reducing potentials. The reducing potential of bee bread extracts was 60% lower than of the bee pollen ones, while that of the honey extract was lower by 54%. On the other hand, the previously published data indicated that the bee bread was characterized by a higher reducing potential than honey (samples collected in the northwest part of Poland) [10]. Moreover, the lowest reducing potential was noted for beeswax (8.14 ± 0.58 µmol Trolox/g). This result was more than 9-times lower compared to the bee pollen. It is noteworthy that the reducing potential depends on the content of bioactive compounds, the harvest period, and the origin of bee products [10].

### 2.4. Antimicrobial Activity

The study results demonstrated that three bee products (bee bread, bee pollen, and honey) possessed antimicrobial activity (Table 4). The highest antimicrobial activity was exhibited by beebread, which inhibited the growth of all the tested bacteria at a concentration of 35% (except *Enterococcus faecalis*, where the inhibiting effect was observed at its 45% concentration; Table 4). Stronger antimicrobial properties of bee bread could be related to a high contribution of gallic acid compared to bee pollen and honey, as well as to the presence of vitexin and orientin, which were not detected in the other bee products tested (Table 2). Ivanišová et al. [33] also showed that bee bread from Ukraine was characterized by antimicrobial activity against Gram-positive (*Bacillus thuringiensis* and *Staphylococcus aureus*), and Gram-negative (*Escherichia coli* and *Salmonella enterica*) strains. Among the other bee products tested, bee pollen also showed antimicrobial activity against all strains used in the study, however, its higher concentrations were needed, i.e., 45–90% against Gram-positive and 35–75% against Gram-negative bacteria. Kacániová et al. [13] also reported the antimicrobial potential of bee pollen from Slovakia towards various strains (e.g., *E. coli*, *S. aureus*, and *Listeria monocytogenes*), while showed that honey inhibited only some of the tested bacteria and at the concentration of 50–90%. Other data also reported the antibacterial activity of Polish multifloral honey against strains of *E. coli* and *S. aureus* [34]. Moreover, honeys’ antimicrobial properties are related to the synergistic effect of hydrogen peroxide and phenolic compounds [15,35]. Bee bread and bee pollen elicited a similar inhibiting effect on the growth of the test strains, however, there was a difference in the diameters of the growth inhibition zones between the samples. This difference may be due to different contents and profiles of bioactive compounds, including mainly polyphenols [33]. Moreover, we can conclude that the bee bread and bee pollen’s antimicrobial activity was stronger against the Gram-negative strains. In the case of honey, a stronger inhibiting effect was observed against the Gram-positive bacteria. The other researchers have also shown that honeys possess stronger antimicrobial activity against Gram-positive than Gram-negative [15].

In our study, beeswax did not show any antimicrobial activity. On the other hand, in the studies presented by Kacániová et al. [13], the wax inhibited the growth of different Gram-negative and Gram-positive strains. This difference may be due to the different methods of extract preparation (70–99.9% methanolic or ethanolic extracts used in the cited study). Moreover, the level of bioactive substances presented in Slovakia’s beeswax could be higher (data not shown) than in the Polish beeswax.

In the next stage of our study, we selected the bee products with the highest antimicrobial properties to determine their MIC, and the results obtained were expressed as % of the concentration of the bee pollen or bee bread which inhibited the growth of the test strains (Table 5). As mentioned above, the bee products with the highest antimicrobial activity turned out to be bee pollen and bee bread. Bee bread showed the antibiotic activity towards *S. aureus* 629G and ATCC29213, *L. monocytogenes* 74 and ATCC1912, and *Salmonella* Typhimurium (in concentration 20%, however, the highest results were obtained against the *Escherichia coli* strain (in concentration 15–20%). Bakour et al. [28] also reported a high antibiotic activity of beebread towards *E. coli*, *S. aureus*, *S.* Typhimurium, and *L. monocytogenes*. While bee pollen was characterized by the higher antibiotic activity only towards the *E. coli* strains (25%). On the other hand, the analyzed samples elicited the weakest inhibiting effect of *S. aureus* G3, *E. faecalis* and *L. monocytogenes* 67 strains (Table 5).

### 2.5. Principal Component Analysis (PCA)

Principal component analysis (PCA) was performed for all samples and variables (TPC, TFC, individual polyphenols, ABTS, DPPH, ACW, ACL, and FRAP–PCA1), as well as between tested strains, TPC, TFC, and individual polyphenols (PCA2) to investigate the correlations between variables and cases. The correlations between the input variables and principal components (PCs) are presented in Figure 2A,B. The first two PCs explained 82.23% and 85.30% of the total data variances for PCA1 and PCA2, respectively. The results obtained in PCA1 (Figure 1—A) demonstrated a strong positive correlation between TPC and DPPH and FRAP assays (r = 0.932 and 0.917, respectively), which indicate a close correlation between the content of phenolic compounds in bee products and their antioxidant and reducing properties. These findings are consistent with a study conducted by Tomczyk et al. [36], who also found a highly positive correlation between the TPC in honey and results of the DPPH and FRAP assays. Moreover, the TPC was positively correlated with results of the ABTS and ACW tests (r = 0.736 and 0.639, respectively). A highly positive correlation between TFC and the DPPH and a positive correlation between TFC and FRAP, ABTS, and ACW were found as well. Furthermore, a strong correlation was determined between ACL test and contents of protocatechuic acid, orientin, vitexin and protocatechuic acid-*O*-hexoside (r = 0.958, 0.925, 0.925, and 0.968, respectively). It suggests that this lipid-soluble antioxidant plays a synergistic role in increasing the antioxidant potential of bee products. Moreover, a positive correlation was detected between contents of gallic, ellagic, neochlorogenic and chlorogenic acids, and antioxidant properties, as well as a negative correlation between antioxidant properties and sakuranetin dimer, caffeic acid, 3,4-di-*O*-caffeoylquinic acid, quercetin, kaempferol, and apigenin contents. This might be because the concentrations of these compounds in bee products are too low to make them play a meaningful antioxidant role.

In PCA2 (Figure 2B), a positive correlation was determined between total phenolic and flavonoid content and microbiological activity as TPC vs. *E. faecalis* 24 (r = 0.630), TPC vs. *E. faecalis* ss1-1 (r = 0.690), TPC vs. *E. coli* ATCC8793 (r = 0.644), and TPC vs. all *S*. Typhimurium bacteria (r = 0.630–0.698). Moreover, a similar observation of relationships was made for microbiological activity and TFC, however, the correlation coefficients were higher and ranged from r = 0.739 (for *E. faecalis* 24 and *S.* Typhimurium 63) to 0.797 (for *S.* Typhimurium). The results obtained indicate that the phenolic and flavonoid compounds present in bee products (mostly beebread and bee pollen) affect the growth and metabolism of bacteria. Moreover, their antimicrobial activity was stronger against Gram-negative strains than the Gram-positive ones. In addition, previously published data also have shown the antimicrobial properties of polyphenolics present in bee products [28,33,34]. The highest positive correlation was achieved between the tested strains and contents of gallic, ellagic, neochlorogenic, chlorogenic, and protocatechuic acids (r = 0.450–0.999). The highest mean value of the “r” coefficient was determined for gallic acid (r = 0.880), followed by ellagic acid (r = 0.846), chlorogenic acid (r = 0.845), neochlorogenic acid (r = 0.743), and protocatechuic acid (r = 0.707). In contrast, a negative correlation was detected between microbial activity and contents of 3,4-di-*O*-caffeoylquinic acid, kaempferol, and apigenin. As with their contribution to the antioxidant activity, these compounds are not present in significant concentrations in bee products to play an important role against growing bacteria. In addition, these three compounds were only present in beeswax, which did not show any antimicrobial properties. Moreover, data obtained shows that phenolic acids rather than flavonoids are the main drivers of the microbial properties of bee products.

The mechanism of the synergistic effect of phytochemicals from bee products needs to be more extensively investigated. Dai, Chen, and Zhou [37] proposed the antioxidant synergistic effect of polyphenols extracted from green tea, which regenerated the antioxidant power of tocopherol, and, therefore, antioxidant activity of tea chemicals was renewed by ascorbic acid. In the next step, the antioxidant and microbial activities of fractionated and isolated polyphenols of each bee product could be compared with the activity of their whole extracts, to study the synergistic effect as it was previously described by Herranz-López et al. [38]. By their study, they proved the synergistic effect of polyphenols extracted from *Hibiscus sabdariffa* against the formation of oxidative species and adipokine secretion [38]. The characteristics of bioactive substances, antioxidant, and antimicrobial properties of the bee products can be helpful for the future research focused on the better utilization of bee products as natural therapeutic agents. The study of Tang et al. [39] on honey/SA/PVA nanofibrous membranes, offers the prospect of using honey as a wound dressing. Moreover, the addition of honey to nanofibers effectively inhibited the growth of both Gram-positive and Gram-negative bacteria. Increasingly advanced research techniques and technologies for the processing and isolation of bioactive compounds also provide opportunities to apply bee products to food as preservatives and agents protecting our health [27]. Thus, there is a need for further research into bee products, their interconnections, and their possible use in medicine, cosmetics, and nutraceuticals.

## 3. Materials and Methods

### 3.1. Chemicals and Reagents

2,2′-Azobis(2-amidonopropane) hydrochloride (AAPH), 2,2′-azinobis(3-ethylbenzothiazoline-6-sulphonic acid) diammonium salt (ABTS), 2,2-diphenyl-1-picrylhydrazyl (DPPH) and 6-hydroxy-2,5,7,8-tetramethylchroman-2-carboxylic acid (Trolox); Folin-Ciocalteu’s phenol reagent; 2,4,6-tris(2-pirydylo)-1,3,5-triazyn (TPTZ); iron (III) chloride hexahydrate; sodium acetate; acetic and hydrochloric acid, gallic acid and quercetin were purchased from Sigma (Sigma Chemical Co., St. Louis, MO, USA). PCL kits for lipophilic (ACL) and hydrophilic antioxidants (ACW) were bought from Analytik Jena (Leipzig, Germany). Reagents of HPLC-MS grade, including acetonitrile, methanol, water, and formic acid, were purchased from Sigma Chemical Co. (St. Louis, MO, USA).

### 3.2. Research Material

Beebread, bee pollen, beeswax and multiflorous honey were obtained from the apiary settled in the Kujawy region (central of Poland) by a professional beekeeper. The bee product samples were collected in the year 2019. Samples were packed in polypropylene bags and kept in refrigeration at 4 °C before analysis.

### 3.3. Extraction of Polyphenols

The bee product samples were extracted according to the method described by Wilczyńska [40] with some modifications. Briefly, 0.5 g of samples were vigorously mixed by 5 min with 5 mL of the methanol/water mixture (80:20, *v/v*) and centrifuged (14,000 RPM, Micro Star 30R, VWR, Radnor, PA, USA) for 10 min at 4 °C. Then extracts have been stored at –20 °C until the analysis.

### 3.4. Identification and Quantification of Polyphenols

The analysis was performed according to the method described by Tomczyk et al. [36]. Briefly, polyphenolic compounds were analyzed using the UPLC−PDA−MS/MS Waters ACQUITY system (Waters, Milford, MA, USA). The polyphenolic detection and identification were based on specific PDA spectra, mass-to-charge ratio, and fragment ions obtained after collision-induced dissociation (CID). The quantitative analysis was based on specific MS transitions in a multiple reaction monitoring (MRM) mode (Table 1). Quantification was achieved by the injection of solutions of known concentrations of phenolic compounds as standards (R ≤ 0.999). All determinations were performed in triplicate and expressed as µg/g.

### 3.5. Determination of Total Phenolics and Flavonoids Content (TPC, TFC)

The measurements of TPC and TF contents were performed in microplates (Infinite M1000 Pro Multimode Microplate Reader; Tecan Männedorf, Switzerland) according to the procedure described previously by Horszwald and Andlauer [41]. The results were calculated as milligram (mg) of gallic acid equivalent (GAE) per gram of sample for TPC, and as mg of quercetin equivalent (QE) per gram for TFC.

### 3.6. Determination of Antioxidant Activity (DPPH, ABTS, and PCL) and Reducing Potential (FRAP)

The DPPH scavenging activity was determined with a method described by Brand-Williams et al. [42]. A decrease in the absorbance of the solution obtained was monitored at 517 nm using an Infinite M1000 PRO plate reader (Tecan Group AG, Männedorf, Switzerland). Results were presented as µmol Trolox per gram of sample. The ABTS test described by Horszwald and Andlauer [41] was used to determine bee product extract’s antioxidant activity. Measurements were carried out using the plate reader. The antioxidant activity was expressed as mmol Trolox/g sample. The PCL method with the Photochem apparatus (Analytik Jena, Leipzig, Germany) was used to measure antioxidants’ effectiveness against superoxide anion radicals. Antioxidant activity was analyzed using the ACW (antioxidative capacities of water-soluble compounds) and ACL (antioxidative capacities of lipid-soluble compounds) kits. The assay was conducted as previously described by Sawicki et al. [10]. The data obtained was presented as µmol Trolox per gram of sample.

The reducing power was determined using the FRAP assay according to Horszwald and Andlauer [41]. The mixture’s absorbance was measured at 593 nm after 5 min reaction using a microplate reader. The FRAP method is based on the reduction of ferric ion by antioxidant compounds.

### 3.7. AntimicrobialActivity

#### 3.7.1. Determination of Antimicrobial Activity by the Well Method

The antimicrobial activity in all the analyzed bee products was determined with the agar well diffusion method as described previously [15]. The test strains originated from a collection of strains maintained at the Department of Industrial and Food Microbiology of the University of Warmia and Mazury in Olsztyn, Poland. Diameters of the inhibition zones of the growth of test Gram-positive and Gram-negative strains induced by the bee bread, bee pollen, beeswax, and honey were identified. Solutions of the analyzed samples were prepared in sterile conical flasks. The following concentrations of the solutions were made: 35, 45, 50, 75, and 90%. Surface cultures (10^4^–10^5^ CFU/mL) of the test strains were started on sterile Petri dishes filled with 20 mL of a nutrient agar growth medium (Merck, Darmstadt, Germany). Next, wells (10 mm in diameter) were made with sterile borer into agar plates containing the bacterial inoculum and filled with the prepared solutions of the analyzed samples, each in an amount of 0.7 mL. The plates were incubated at 37 °C for 24 h. After the incubation, the diameters of the inhibition zones of the growth of the test strains around the wells were determined. The experiment was performed in triplicate.

#### 3.7.2. Determination of Minimum Inhibitory Concentration (MIC)

The MIC, i.e., the lowest concentration of honey inhibiting the growth of a test strain, was identified in the bee products which showed the highest antimicrobial activity [15]. The test was made by performing subsequent dilutions (the tested ranges were 100%, 50%, 25%, 12.5%, 6.25%, etc.) of bee bread and bee pollen in a liquid stock medium (Antibiotic Medium Broth, Merck, Darmstadt, Germany). 1 mL of the growth medium was transferred to each of the test tubes, which were then inoculated with a 24 h culture of the test strains with 10^4^ cells/mL inoculum in an amount of 0.1 mL. The samples were incubated for 24 h at 37 °C. After incubation, it was determined whether there was an increase in the test strain by plating the cultures by the surface method on selective media for the given test strains (TBX for *E. coli*, Slanetz’a Bartley’a for *Enterococcus sp*., XLD for *Salmonella sp*., Baird Parker’a for *S. aureus*, and ALOA for *Listeria monocytogenes*).

### 3.8. Statistical Analysis

The data are presented as mean values ± standard deviations of triplicate measurement. The differences between samples were analyzed by a one-way ANOVA with LSD Fisher’s post hoc test (*p* < 0.05). The Pearson correlation test for correlation analysis was used. Furthermore, data related to antioxidant properties, microbial activity (90% of sample concentration was used), contents of individual polyphenolic compounds, TPC, and TFC were subjected to principal component analysis (PCA). The statistical analysis was performed using STATISTICA 13.0 software (StatSoft Inc., Tulsa, OK, USA).

## 4. Conclusions

As previously mentioned, this is the first study that presents the composition of polyphenolics in four different bee products obtained from the same batch. The TPC and TFC, and concentrations of individual polyphenolics varied significantly among the bee products tested. The highest level of bioactive substances was noted in the bee pollen compared to the bee bread, honey, and beeswax. Furthermore, the polyphenols present in bee pollen were found to be the major contributors to its high antioxidant activity. Moreover, each bee product was characterized by specific antimicrobial properties correlated with TPC, TFC, and individual polyphenolic content. Principal component analysis (PCA) results showed that the antioxidant activity determined by DPPH and ABTS assays might be related to (the synergistic effect) the content of ellagic, neochlorogenic and chlorogenic acids, and pinobanksin. Whereas, protocatechuic acid-*O*-hexoside and protocatechuic acid may be attributable to the combined/synergistic effect of antioxidant activity determined by the ACL assay, while gallic acid may be responsible for the antioxidant activity determined by the ACW assay. The reducing potential (FRAP assay) of the samples may be attributable to the combined/synergistic effect of quercetin 3-*O*-glucuronide as well as TPC. On the other hand, the same statistical analysis showed that the antimicrobial properties of the honey, bee pollen, and bee bread may be related to the synergistic effect of gallic acid, ellagic acid, chlorogenic acid, protocatechuic acid, orientin, vitexin, and protocatechuic acid-*O*-hexoside. Thus, bee products, especially bee pollen and bread, could be recommended to be attractive food additives to enhance food rich in bioactive components and with the possibility to increase products’ functional properties.

## Figures and Tables

**Figure 1 molecules-27-01301-f001:**
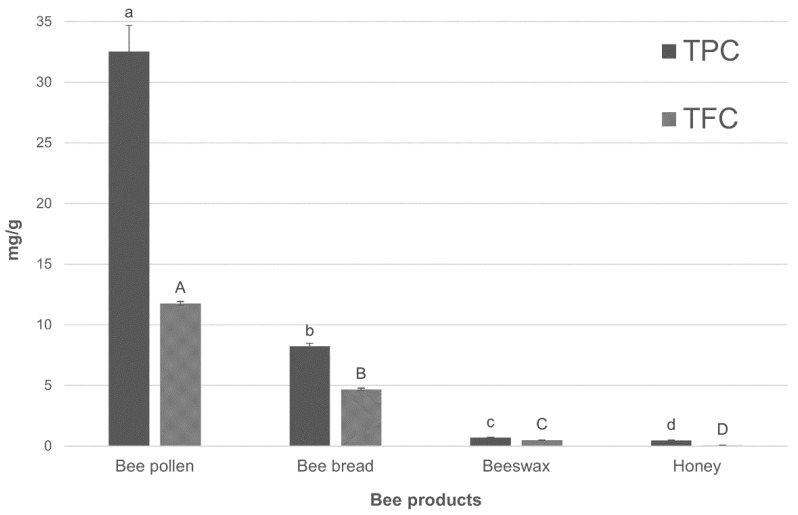
Total phenolic content (TPC) and total flavonoid content (TFC) in bee products. TPC is expressed as mg of gallic acid equivalent (GAE) per gram of sample, while, TF is expressed as mg of quercetin equivalent (QE) per gram. The results for each sample are reported as the mean value of three repetitions. a–d: different letters indicate significant differences of TPC values (*p* < 0.05); A–D: different letters indicate significant differences of TFC values (*p* < 0.05).

**Figure 2 molecules-27-01301-f002:**
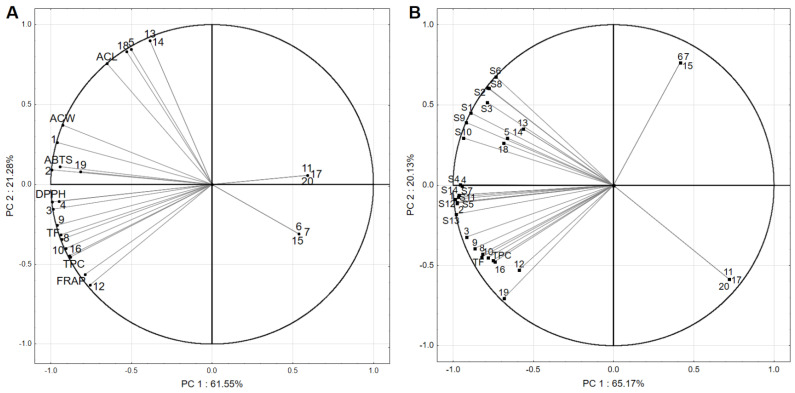
Principal component analysis of TPC, TFC, individual polyphenols, and antioxidant properties (ABTS, DPPH, ACW, ACL, and FRAP) (**A**), and between tested strains, TPC, TFC, and individual polyphenols (**B**) (1–20 represent the number of polyphenols identified in the bee products Table 2, while S1–S14 represent the strain used in the study—Table 5).

**Table 1 molecules-27-01301-t001:** Polyphenolic compounds detected in the bee products using UPLC-PDA-ESI-MS.

No.	Compound	R_t_ [min]	MS [*m*/*z*]	MS/MS [*m*/*z*]	λ_max_ [nm]	Sample
1	gallic acid	1.12	169	125	271	H, BB, BP
2	ellagic acid	1.16	301	283/200/175	267	H, BB, BP
3	neochlorogenic acid	1.71	353	191/179	262	H, BB, BP
4	chlorogenic acid	2.03	353	191/179	281	H, BB, BP
5	protocatechuic acid	2.23	153	109	291	H, BB, BP
6	sakuranetin dimer	3.19	551	285/179/164	307	H
7	caffeic acid	3.96	179	135	300	H
8	rutin	4.44	609	301	286/338	BB, BP
9	sinapic acid	4.73	223	175/164	295	H, BB, BP
10	isorhamnetine 3-*O*-rutinoside	4.76	623	315/314	266, 309	BB, BP
11	3,4-di-*O*-caffeoylquinic acid	5.09	515	179/191	-	W
12	quercetin 3-*O*-glucuronide	5.35	477	301	269, 324	BB, BP
13	orientin	5.39	447	357/339	265, 316	BB
14	vitexin	6.12	431	341/311	264, 315	BB
15	quercetin	6.45	301	179/151	256/355	H
16	epicatechin	7.44	289	245/203	319	BB, BP
17	kaempferol	7.53	285	257/201/185	286	W
18	protocatechuic acid- *O*-hexoside	8.15	315	153	287	BB, BP, W
19	pinobanksin	8.24	271	185/151	290	BB, BP, W
20	apigenin	8.65	269	179/225	338/ 346	W

Abbreviation: Rt—retention time; H—honey; BP—bee pollen; BB—bee bread; and W—beeswax.

**Table 2 molecules-27-01301-t002:** Contribution of individual polyphenolic compounds in the bee products.

No.	Compound	Bee Product			
		Bee Pollen	Beebread	Honey	Beeswax
1	gallic acid	21.3	32.6	69.2	ND
2	ellagic acid	2.2	2.6	6.6	ND
3	neochlorogenic acid	0.9	0.7	1.5	ND
4	chlorogenic acid	0.7	0.8	13.2	ND
5	protocatechuic acid	0.1	0.8	0.2	ND
6	sakuranetin dimer	ND	ND	4.6	ND
7	caffeic acid	ND	ND	0.8	ND
8	rutin	10.0	5.1	ND	ND
9	sinapic acid	42.8	27.3	1.8	ND
10	isorhamnetine 3-*O*-rutinoside	5.8	2.5	ND	ND
11	3.4-di-*O*-caffeoylquinic acid	ND	ND	ND	1.9
12	quercetin 3-*O*-glucuronide	7.9	0.3	ND	ND
13	orientin	ND	7.1	ND	ND
14	vitexin	ND	15.2	ND	ND
15	quercetin	ND	ND	2.2	ND
16	epicatechin	7.2	2.6	ND	ND
17	kaempferol	ND	ND	ND	6.0
18	protocatechuic acid-*O*-hexoside	0.2	1.6	ND	3.4
19	pinobanksin	0.7	0.8	ND	85.7
20	apigenin	ND	ND	ND	3.0
	**Total [µg/g]**	**79.39 ± 0.31 ^a^**	**56.27 ± 0.89 ^b^**	**1.61 ± 0.02 ^c^**	**0.39 ± 0.00 ^d^**

The results for each sample are reported as the mean value of 3 repetitions. a–d: different letters indicate significant differences (*p* < 0.05) within the lines.

**Table 3 molecules-27-01301-t003:** Antioxidant activity (determined using PCL, DPPH, and ABTS assays) and reducing potential (FRAP assay) of bee products.

Parameter/Sample	PCL [µmol Trolox/g]		DPPH [µmol Trolox/g]	ABTS [mmol Trolox/g]	FRAP [µmol Trolox/g]
	ACL	ACW			
Bee pollen	410.13 ± 19.56 ^b^	129.29 ± 2.75 ^b^	16.97 ± 1.19 ^a^	32.56 ± 0.30 ^a^	76.94 ± 4.48 ^a^
Bee bread	1017.83 ± 56.03 ^a^	162.16 ± 2.83 ^a^	10.26 ± 2.27 ^b^	31.60 ± 0.16 ^b^	31.23 ± 1.96 ^b^
Beeswax	210.59 ± 0.41 ^c^	4.72 ± 0.04 ^c^	0.53 ± 0.02 ^c^	5.96 ± 0.05 ^d^	8.14 ± 0.58 ^c^
Honey	1.53 ± 0.02 ^d^	4.72 ± 0.17 ^c^	0.18 ± 0.04 ^d^	15.68 ± 0.60 ^c^	35.36 ± 1.03 ^b^

The results for each sample are reported as the mean value of 3 repetitions. a–d: different letters indicate significant differences (*p* < 0.05) within the column.

**Table 4 molecules-27-01301-t004:** Diameters of the growth inhibition zones for the Gram-positive and Gram-negative strains formed by the bee products.

Test Strain	Sample Concentration [%]	Diameters of the Growth Inhibition Zones [mm]			
		Honey	Bee Pollen	Bee Bread	Beeswax
Gram-positive strains					
*Staphylococcus aureus* G3	90	12.0 ± 0.0 ^c^	16.0 ± 1.0 ^b^	22.0 ± 2.0 ^a^	0
	75	0	14.0 ± 0.0	18.0 ± 2.0	0
	50	0	12.0 ± 0.0	16.0 ±1.0	0
	45	0	12.0 ± 0.0	14.0 ±1.0	0
	35	0	0	12.0 ± 0.0	0
*Staphylococcus aureus* 629G	90	16.0 ± 1.0	16.0 ± 1.0	20.0 ± 2.0	0
	75	14.0 ± 1.0	14.0 ± 1.0	18.0 ± 2.0	0
	50	12.0 ± 0.0 ^b^	12.0 ± 0.0 ^b^	18.0 ± 1.0 ^a^	0
	45	0	0	16.0 ± 1.0	0
	35	0	0	12.0 ± 0.0	0
*Staphylococcus aureus* ATCC29213	90	16.0 ± 1.0	18.0 ± 2.0	18.0 ± 1.0	0
	75	14.0 ± 1.0	14.0 ± 1.0	16.0 ± 1.0	0
	50	12.0 ± 0.0 ^b^	14.0 ± 1.0 ^a^	14.0 ± 1.0 ^a^	0
	45	0	12.0 ± 0.0	12.0 ± 0.0	0
	35	0	0	12.0 ± 0.0	0
*Enterococcus faecalis* 24	90	0	14.0 ± 1.0	18.0 ± 1.0	0
	75	0	12.0 ± 0.0	16.0 ± 1.0	0
	50	0	0	14.0 ± 1.0	0
	45	0	0	12.0 ± 0.0	0
	35	0	0	0	0
*Enterococcus faecalis* ss1-1	90	0	14.0 ± 1.0	16.0 ± 1.0	0
	75	0	12.0 ± 0.0	16.0 ± 1.0	0
	50	0	0	14.0 ± 1.0	0
	45	0	0	12.0 ± 0.0	0
	35	0	0	0	0
*Listeria monocytogenes* 67	90	14.0 ± 1.0 ^b^	12.0 ± 0.0 ^c^	20.0 ± 2.0 ^a^	0
	75	0	0	16.0 ± 1.0	0
	50	0	0	14.0 ± 1.0	0
	45	0	0	12.0 ± 0.0	0
	35	0	0	12.0 ± 0.0	0
*Listeria monocytogenes* 74	90	0	12.0 ± 0.0	18.0 ± 2.0	0
	75	0	12.0 ± 0.0	16.0 ± 1.0	0
	50	0	0	14.0 ± 1.0	0
	45	0	0	12.0 ± 0.0	0
	35	0	0	12.0 ± 0.0	0
*Listeria monocytogenes* ATCC1912	90	12.0 ± 0.0 ^b^	12.0 ± 0.0 ^b^	16.0 ± 1.0 ^a^	0
	75	0	0	14.0 ± 1.0	0
	50	0	0	12.0 ± 0.0	0
	45	0	0	12.0 ± 0.0	0
	35	0	0	12.0 ± 0.0	0
Gram-negative strains					
*Escherichia coli* 14169	90	12.0 ± 0.0 ^c^	18.0 ± 2.0 ^b^	24.0 ± 2.0 ^a^	0
	75	0	16.0 ± 2.0	20.0 ± 2.0	0
	50	0	14.0 ± 1.0	20.0 ± 2.0	0
	45	0	14.0 ± 1.0	18.0 ± 1.0	0
	35	0	12.0 ± 0.0	14.0 ±1.0	0
*Escherichia coli* 25922	90	12.0 ± 0.0 ^b^	20.0 ± 2.0 ^a^	20.0 ± 2.0 ^a^	0
	75	0	18.0 ± 2.0	20.0 ± 2.0	0
	50	0	14.0 ± 1.0	18.0 ± 1.0	0
	45	0	14.0 ± 1.0	16.0 ± 1.0	0
	35	0	12.0 ± 0.0	14.0 ± 1.0	0
*Escherichia coli* ATCC8793	90	0	16.0 ± 2.0	20.0 ± 2.0	0
	75	0	16.0 ±1.0	18.0 ± 2.0	0
	50	0	14.0 ± 1.0	16.0 ± 1.0	0
	45	0	12.0 ± 0.0	14.0 ± 1.0	0
	35	0	12.0 ± 0.0	14.0 ± 1.0	0
*Salmonella* Typhimurium	90	0	16.0 ± 1.0	18.0 ± 1.0	0
	75	0	14.0 ± 1.0	18.0 ± 1.0	0
	50	0	12.0 ± 0.0	16.0 ± 1.0	0
	45	0	12.0 ± 0.0	14.0 ± 1.0	0
	35	0	12.0 ± 0.0	12.0 ± 0.0	0
*Salmonella* Typhimurium 235	90	0	14.0 ± 1.0	16.0 ± 1.0	0
	75	0	12.0 ± 0.0	16.0 ± 1.0	0
	50	0	12.0 ± 0.0	14.0 ± 1.0	0
	45	0	0	12.0 ± 0.0	0
	35	0	0	12.0 ± 0.0	0
*Salmonella* Typhimurium 63	90	0	14.0 ± 1.0	18.0 ± 2.0	0
	75	0	12.0 ± 0.0	18.0 ± 1.0	0
	50	0	0	16.0 ± 1.0	0
	45	0	0	14.0 ± 0.0	0
	35	0	0	12.0 ± 0.0	0

Data are a mean ± standard deviation (*n* = 3). Statistical analysis was performed for bee products concentration at which at least three samples demonstrated antimicrobial activity. Different letters in the same row indicate statistical significance (*p* < 0.05).

**Table 5 molecules-27-01301-t005:** MIC of the bee bread and bee pollen (%, *v/v*).

No.	Test Strain	Sample	
		Bee Pollen	Bee Bread
S1	*Staphylococcus aureus* G3	50.0	50.0
S2	*Staphylococcus aureus* 629G	50.0	25.0
S3	*Staphylococcus aureus* ATCC29213	50.0	25.0
S4	*Enterococcus faecalis* 24	50.0	50.0
S5	*Enterococcus faecalis* ss1-1	50.0	50.0
S6	*Listeria monocytogenes* 67	50.0	50.0
S7	*Listeria monocytogenes* 74	50.0	25.0
S8	*Listeria monocytogenes* ATCC1912	50.0	25.0
S9	*Escherichia coli* 14169	25.0	15.0
S10	*Escherichia coli* 25922	25.0	15.0
S11	*Escherichia coli* ATCC8793	25.0	20.0
S12	*Salmonella* Typhimurium	50.0	25.0
S13	*Salmonella* Typhimurium 235	50.0	25.0
S14	*Salmonella* Typhimurium 63	50.0	25.0

## Data Availability

Not applicable.

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
