# Peer review of "The Profile of Polyphenolic Compounds, Contents of Total Phenolics and Flavonoids, and Antioxidant and Antimicrobial Properties of Bee Products"

_molecules, 2022, doi:10.3390/molecules27041301_

Round 1

Reviewer 1 Report

This study aims to be the first to characterise natural products from different bee products from the same hives. I have reviewed this paper in its first submission. Clear improvements can be seen in this new version. My review and suggestions are below:

1) This time the authors re-worded the identification of 20 compounds accurately (line 18/19) and the use of tables (Tables 1 and 2) have improved how information is organised. I believe that a chromatogram of  Table 1 could be as a supplementary figure?  I believe it should be clarified if the compounds identified in Table 1 are the ones identified in higher quantities?

2) Regarding my previous comments on the "vigorous mixing", I still believe this is not a very accurate extraction method. If quantification of natural products are done, we would need more information than this. For example, if the authors in this paper mix for 1 minute and authors from other paper follow exactly the same procedure but mix for 10 minutes, will we have the same concentration of extract? Will the same dilutions give exactly the same activity? I believe they wouldn't. Even while comparing results between bee products in this paper, the poor extraction method may yield differently concentrated products, and make the NP quantification and activity invalid.

I would be more than happy to endorse this paper if these two points are addressed.

Author Response

Itemized list of changes addressing Editor and Reviewer comments and correction checklist.

This study aims to be the first to characterise natural products from different bee products from the same hives. I have reviewed this paper in its first submission. Clear improvements can be seen in this new version. My review and suggestions are below:

Our answer: We are very grateful for this supportive comment and appreciate all suggestions and revision recommendations. We have gone over all the points that you have raised and we believe that the applied changes have increased the scientific value of our manuscript.

1) This time the authors reworded the identification of 20 compounds accurately (line 18/19) and the use of tables (Tables 1 and 2) have improved how information is organised. I believe that a chromatogram of  Table 1 could be as a supplementary figure?  I believe it should be clarified if the compounds identified in Table 1 are the ones identified in higher quantities?

Our answer: Thank you for a very positive assessment of our study. Since there is a very big difference in concentration (the size of peak) between polyphenolic compounds found in bee products therefore a range of smaller peaks would be invisible on chromatograms. Only major peaks will be dominated on chromatograms. The profile of individual polyphenolics in the bee products was presented in Table 2.

2) Regarding my previous comments on the "vigorous mixing", I still believe this is not a very accurate extraction method. If quantification of natural products are done, we would need more information than this. For example, if the authors in this paper mix for 1 minute and authors from other paper follow exactly the same procedure but mix for 10 minutes, will we have the same concentration of extract? Will the same dilutions give exactly the same activity? I believe they wouldn't. Even while comparing results between bee products in this paper, the poor extraction method may yield differently concentrated products, and make the NP quantification and activity invalid.

Our answer: Thank you for a very insightful review and detailed suggestions. We agree with your opinion, and according to your suggestion, we added information on extraction time and we believe that it improved the clarity of our manuscript.

Reviewer 2 Report

Inconsistence in the usage of TPC and TFC

Fig-1: No clarity on alphabets on small and big letters

GAE/g is expressed as mg / µg / ng ?? while QE/g is expressed as mg

Author Response

Itemized list of changes addressing Editor and Reviewer comments and correction checklist.

Inconsistence in the usage of TPC and TFC

Our answer: Thank you for this comment. The correction was done.

Fig-1: No clarity on alphabets on small and big letters

Our answer: Thank you for this comment. The correction was done. Sections of paper addressing to the above comment: manuscript, page 3, Figure 1.

GAE/g is expressed as mg / µg / ng ?? while QE/g is expressed as mg

Our answer: Thank you for this comment. It was our mistype. The correction was done. Sections of paper addressing to the above comment: manuscript, page 3, Figure 1.

Reviewer 3 Report

The manuscript entitled “The profile of polyphenolic compounds, contents of total phe-nolics and flavonoids, and antioxidant and antimicrobial properties of bee products” have described composition, antioxidants activity and antimicrobial properties of bee products.

The study was written carefully and well in terms of language. Authors should correct manuscript according to the suggestion.

Minor issues:

Line 530 the range of tested concentration should be added

Line 566 - 585: This part should be moved to the part of the discussion

Table 3 in case of ABTS it should be mmol or µmol Trolox/g

Author Response

Itemized list of changes addressing Editor and Reviewer comments and correction checklist.

The manuscript entitled “The profile of polyphenolic compounds, contents of total phenolics and flavonoids, and antioxidant and antimicrobial properties of bee products” have described composition, antioxidants activity and antimicrobial properties of bee products.

The study was written carefully and well in terms of language. Authors should correct manuscript according to the suggestion.

Our answer: Thank you for a very positive assessment of our study.

Minor issues:

Line 530 the range of tested concentration should be added

Our answer: We agree with the comment. The additional information was added. Sections of paper addressing to the above comment: manuscript, page 16, lines 548-549.

Line 566 - 585: This part should be moved to the part of the discussion.

Our answer: We agree with the comment. The correction was done. Sections of paper addressing to the above comment: manuscript, page 13-14, lines 436-454.

Table 3 in case of ABTS it should be mmol or µmol Trolox/g

Our answer: Thank you for this comment. The correction was done. Sections of paper addressing to the above comment: manuscript, page 8, Table 3.

Round 2

Reviewer 1 Report

The author's corrections and comments are very adequate. The scientific soundness of this paper has been constantly improved.

This manuscript is a resubmission of an earlier submission. The following is a list of the peer review reports and author responses from that submission.

Round 1

Reviewer 1 Report

This study aims to be the first to characterise natural products from different bee products from the same hives. Some suggestions are below:

Separation and identification of compounds. How were the compounds identified on UPLC-MS?

"The bee products contained twenty polyphenol", authors say. I would assume this is not true as the identification was "achieved by (...) [using] known (...) phenolic compounds as standards"? Surely bee products would contain more than 20 polyphenols, this study only identified 20 because they were using 20 standards?

Antimicrobial activity and compounds extraction: why was the extraction set at 4C? I would believe that, if done at room temperature (not disregarding the environment where the samples are naturally), you could have achieved better yields and possibly converge results with other studies?

What was the yield for crude extracts after centrifugation?

MICs would be more accurate in concentrations (mg/mL, for example). I would suggest that Table 5 could present results of MICs in concentrations.

Minor/editing issues:

  • Line 15, typo with commas.
  • Line 43, in vitro italic.
  • Line 43, dash missing.
  • Line 54, kinds to types.
  • .....
  • ....

This manuscript requires extensive editing regarding English language. My list of typos suggested for review is quite extensive. Manuscript should be edited my a native English speaker.

I am afraid this paper may not be showing the full potential of bee products by choosing a extraction method that only offers low yield, by identifying target natural products (and not the whole lot), and by not pretending MIC as a concentration, so it can be compared with other studies.

Author Response

Dear Reviewer 1

I have presented below answer to your comments in the following schema:

This study aims to be the first to characterise natural products from different bee products from the same hives. Some suggestions are below: 

Separation and identification of compounds. How were the compounds identified on UPLC-MS? "The bee products contained twenty polyphenol", authors say. I would assume this is not true as the identification was "achieved by (...) [using] known (...) phenolic compounds as standards"? Surely bee products would contain more than 20 polyphenols, this study only identified 20 because they were using 20 standards?

Our answer: UPLC−PDA−MS/MS method was used for polyphenol analysis (sections of paper addressed to the above comment: manuscript, page 14, line: 501-509). The method is based on analysing parent ions and their fragments, which gives almost 100% certainty of the identified relationships. Additionally, the identified compounds were confirmed by standards used to plot standard curves and calculate the concentrations of the identified compounds. The Reviewer raises an interesting concern about the numbers of identified compounds. Some of the compounds present in the tested material may be present in trace amounts, making it impossible to identify a larger number of compounds. Nevertheless, it is worth mentioning that this work presents many identified compounds that have not been presented so far in scientific research on bee products.

Antimicrobial activity and compounds extraction: why was the extraction set at 4C? I would believe that, if done at room temperature (not disregarding the environment where the samples are naturally), you could have achieved better yields and possibly converge results with other studies?

Thank you for this comment. The extraction was carried out at room temperature according to the rules mentioned by the Reviewer. However, the samples after extraction were centrifuged at 4 °C to prevent degradation of compounds by increasing temperature during centrifugation of samples.

MICs would be more accurate in concentrations (mg/mL, for example). I would suggest that Table 5 could present results of MICs in concentrations.

The reviewer raises an interesting concern. In the available literature, results of this type are presented in the manuscript. We also decided to present MIC data in % concentration. This was mainly due to: comparing our results with those of other authors and increasing the availability and citation of our article.

Minor/editing issues:

  • Line 15, typo with commas.
  • Line 43, in vitro italic.
  • Line 43, dash missing.
  • Line 54, kinds to types.

Thank you for this comment. It was our mistype. The correction was done.

This manuscript requires extensive editing regarding English language. My list of typos suggested for review is quite extensive. Manuscript should be edited my a native English speaker.

English was checked and all mistakes verified.

Reviewer 2 Report

This study provides useful information about the profile of phenolic compounds, total phenolic and total flavonoid contents as well as antioxidant and antimicrobial activities of several bee products. However, many simple mistakes and typographical errors were found in this manuscript. Please consider minor points (red color indicated in the attached PDF file) that need to be corrected at least. In addition, there are also some grammatical mistakes in the main text. I encourage the authors to have your paper checked by native speakers or professional grammatical editing services. I hope my comments are useful for the improvement of this paper.

Author Response

Dear Reviewer 2

This study provides useful information about the profile of phenolic compounds, total phenolic and total flavonoid contents as well as antioxidant and antimicrobial activities of several bee products. However, many simple mistakes and typographical errors were found in this manuscript. Please consider minor points (red color indicated in the attached PDF file) that need to be corrected at least. In addition, there are also some grammatical mistakes in the main text. I encourage the authors to have your paper checked by native speakers or professional grammatical editing services. I hope my comments are useful for the improvement of this paper.

Our answer: Thank you for a very positive assessment of our study. We agree with the comment. The correction was done.

Reviewer 3 Report

The authors presented a study entitle “The profile of polyphenolic compounds, total phenolics and flavonoids contents, antioxidant and antimicrobial properties of bee products”. This study provides interesting data on the phenolic compound profile and antioxidant and antibacterial properties of bee bread, bee pollen, beeswax and honey. The authors conducted a meticulous study describing in detail the methods used and the results obtained. The purpose of the work is clear and the results provide insightful perspectives for future research on the functional value of bee products.

However this reviewer has some suggestions that could increase the scientific community interest and improve the quality of the manuscript.

In detail:

at line 130 the authors summarize that the different level of phenolic and flavonoid compounds found in bee products can result from three main factors: extraction method, botanical origin and harvest time. It would be better to describe how these three factors affect the content of phenolic compounds and flavonoids.

In addition, it is already known that the beneficial properties of many natural products derive from the complex mixture of antioxidant molecules and not from the activity of the single molecule. Although the authors mention that the antioxidant and antimicrobial activity is due to the synergistic effect of the molecules studied, it would be appropriate to give more emphasis to this aspect.  The authors should describe how substances such as polyphenols work in combination to counteract oxidative stress. This description could better clarify the importance of bee products and in particular those with a higher flavonoid content.

Furthermore, based on the results obtained by the authors, a more in-depth discussion regarding the future developments and applications of bee products in the medical, cosmetic and nutraceutical fields should be added.

At last, a complete revision of the text is recommended to carefully correct typo.

Author Response

Dear Reviewer 3

The authors presented a study entitle “The profile of polyphenolic compounds, total phenolics and flavonoids contents, antioxidant and antimicrobial properties of bee products”. This study provides interesting data on the phenolic compound profile and antioxidant and antibacterial properties of bee bread, bee pollen, beeswax and honey. The authors conducted a meticulous study describing in detail the methods used and the results obtained. The purpose of the work is clear and the results provide insightful perspectives for future research on the functional value of bee products.

However this reviewer has some suggestions that could increase the scientific community interest and improve the quality of the manuscript.

Our answer: Thank you for a very positive assessment of our study.

In detail: at line 130 the authors summarize that the different level of phenolic and flavonoid compounds found in bee products can result from three main factors: extraction method, botanical origin and harvest time. It would be better to describe how these three factors affect the content of phenolic compounds and flavonoids.

We agree with the comment. The additional information was added

Sections of paper addressing to the above comment: manuscript, page 4, lines 135-142.

In addition, it is already known that the beneficial properties of many natural products derive from the complex mixture of antioxidant molecules and not from the activity of the single molecule. Although the authors mention that the antioxidant and antimicrobial activity is due to the synergistic effect of the molecules studied, it would be appropriate to give more emphasis to this aspect.  The authors should describe how substances such as polyphenols work in combination to counteract oxidative stress. This description could better clarify the importance of bee products and in particular those with a higher flavonoid content.

We agree with the comment. The additional information was added

Sections of paper addressing to the above comment: manuscript, page 16, lines 601-610.

Furthermore, based on the results obtained by the authors, a more in-depth discussion regarding the future developments and applications of bee products in the medical, cosmetic and nutraceutical fields should be added.

We agree with the comment. The additional information was added.

Sections of paper addressing to the above comment: manuscript, page 16, lines 610-619.

Round 2

Reviewer 1 Report

This review took into account the authors corrections and comments to my previous suggestions. Many thanks. My comments are below yours, in red.

Separation and identification of compounds. How were the compounds identified on UPLC-MS? "The bee products contained twenty polyphenol", authors say. I would assume this is not true as the identification was "achieved by (...) [using] known (...) phenolic compounds as standards"? Surely bee products would contain more than 20 polyphenols, this study only identified 20 because they were using 20 standards?

Our answer: UPLC−PDA−MS/MS method was used for polyphenol analysis (sections of paper addressed to the above comment: manuscript, page 14, line: 501-509). The method is based on analysing parent ions and their fragments, which gives almost 100% certainty of the identified relationships. Additionally, the identified compounds were confirmed by standards used to plot standard curves and calculate the concentrations of the identified compounds. The Reviewer raises an interesting concern about the numbers of identified compounds. Some of the compounds present in the tested material may be present in trace amounts, making it impossible to identify a larger number of compounds. Nevertheless, it is worth mentioning that this work presents many identified compounds that have not been presented so far in scientific research on bee products.

Dear authors many thanks for your reply. I am afraid the identification done via tandem MS is a level 3 "tentative identification", far from the 100% certainty referred. What gives that high certainty is the level 5 "unique feature" achieved using your standards. While the identification of peaks of your chromatogram done via standards is absolutely correct, my question was regarding the wording in your paper. "The bee products contained twenty polyphenols" is wrong, as something like 'Twenty products were identified in bee products' sounds more accurate.

In your reply you have focused some attention to the number of compounds identified. This was not a problem in my first review. The only issue was how the authors choose to write about their identification. To put it simply, did the authors only find those 20 compounds because they were the existing standards that were in the laboratory, or were they found because they were the 20 more significant compounds, with larger AUC for example? 

Antimicrobial activity and compounds extraction: why was the extraction set at 4C? I would believe that, if done at room temperature (not disregarding the environment where the samples are naturally), you could have achieved better yields and possibly converge results with other studies?

Thank you for this comment. The extraction was carried out at room temperature according to the rules mentioned by the Reviewer. However, the samples after extraction were centrifuged at 4 °C to prevent degradation of compounds by increasing temperature during centrifugation of samples.

Many thanks for this comment. I assume the "vigorous mixing" is the extraction method. If this is correct, this is a very poorly extraction method/description as the natural products were not extracted to its full potential. This can be seen in multiple other articles regarding the extraction of natural products from honey, for example. It is acceptable to compare extracts obtained from the same method, as the authors did in this paper, but I believe a comment about the poor extent of the extraction method should be added to the paper. This is because ideally, if other researchers are looking to studying the antimicrobial activity of natural products from the same sources, they should not use the method here described as it is by far not the method that presents best yields.

MICs would be more accurate in concentrations (mg/mL, for example). I would suggest that Table 5 could present results of MICs in concentrations.

The reviewer raises an interesting concern. In the available literature, results of this type are presented in the manuscript. We also decided to present MIC data in % concentration. This was mainly due to: comparing our results with those of other authors and increasing the availability and citation of our article.

Many thanks for your comment. I would have to strongly disagree with this. Only an MIC (minimum inhibitory concentration) in mass/volume would accurately assess antimicrobial activity (see CLSI standards). At least to the extent of a journal with this impact factor. Let me say the literature you refer (citations 15 and 40) are very poor and do not represent the standards for good antimicrobial activity reporting based on broth dilution.

It should be clear to the authors that this percentage is the dilution of the extract, which will be correlated to the amount extracted in the beginning. It is not right to assume all other papers will have exactly the same starting material concentration so we are able to compare percentages at the end. Also all the issues inherent to the poor extraction method will ultimately give biased percentages in the results. For example, if the authors in this paper mix for 1 minute and authors from other paper follow exactly the same procedure but mix for 10 minutes, will we have the same concentration of extract? Will the same dilutions give exactly the same activity? Will the same "MIC%" give the same result? - No, they won't. That is why we have standards for establishing the MIC for a drug or group of drugs.

It is with regret I have to say that this paper lacks accuracy that is key to communicate sound scientific articles. I should reassure to the authors of this paper that the "increased availability and citations" of your paper should always be based on the most accurate methods of obtain the answers for the problem that you aim to answer.